# Don't lie to your friends:
# Learning what you know from collaborative self-play

**Jacob Eisenstein**      **Reza Aghajani**      **Adam Fisch**      **Dheeru Dua**

**Fantine Huot**      **Mirella Lapata**      **Vicky Zayats**      **Jonathan Berant** [*]

Google DeepMind
jeisenstein@google.com

## Abstract

To be helpful assistants, AI agents must be aware of their own capabilities and limitations. This includes knowing when to answer from parametric knowledge versus using tools, when to trust tool outputs, and when to abstain or hedge. Such capabilities are hard to teach through supervised fine-tuning because they require constructing examples that reflect the agent's specific capabilities. We therefore propose a radically new approach to teaching agents what they know: *collaborative self-play*. We construct multi-agent collaborations in which the group is rewarded for collectively arriving at correct answers. The desired meta-knowledge emerges from the incentives built into the structure of the interaction. We focus on small societies of agents that have access to heterogeneous tools (corpus-specific retrieval), and therefore must collaborate to maximize their success with minimal effort. Experiments show that group-level rewards for multi-agent communities can induce policies that *transfer* to improve tool use and selective prediction in single-agent scenarios.

## 1 Introduction

While conversational assistants based on language models (LMs) are having unprecedented success (Gemini Team, 2024; Llama Team, 2024; OpenAI, 2024; DeepSeek-AI, 2025), there is growing evidence that skills that are crucial for successful human-AI collaboration are still lacking. Compared with human speakers, LMs perform fewer grounding actions, such as asking clarification questions (Shaikh et al., 2024); they do not express uncertainty faithfully (Zhou et al., 2024; Yona et al., 2024; Stengel-Eskin et al., 2024); and they often use outputs from external tools incorrectly (Yoran et al., 2024; Wu et al., 2024).

People build and use such skills when communicating in natural language to achieve collaborative goals (Grice, 1975). Such collaborations are more likely to be successful when participants are truthful about their knowledge, ask for help when needed, and use external resources when their knowledge is limited. This requires meta-cognitive capabilities such as estimating and communicating uncertainty, and learning to evaluate the reliability of provided information. But post-training procedures for endowing these skills onto LMs are devoid of this social, goal-oriented signal, relying instead on supervised fine-tuning on curated examples (Zhang et al., 2023; Stengel-Eskin et al., 2024; Li et al., 2023; Yoran et al., 2024). This also entails that data collection needs to be carried out per skill.

In this work, we propose *collaborative self-play* (CSP) as a mechanism to teach language models to be more helpful, by constructing multi-agent environments where accomplishing goals relies on learning to be an efficient and effective communicator. Specifically, a small

---

[*]Core contributors: JE, RA, AF, JB.

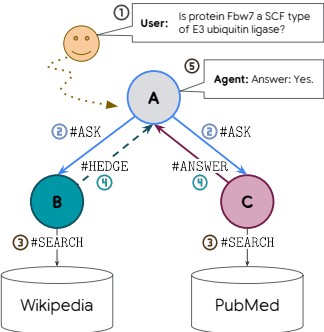

Figure 1: **Collaborative self-play for question answering.** A small society of agents is asked to answer a question. Success requires cooperation: agents must use their unique resources and share not only information but also calibrated expressions of confidence. Here, Agent A receives the initial query from the user ①, and asks for help from Agent B and C ②, who search ③ and then respond to Agent A ④, after which Agent A chooses a final response ⑤. Agent B does not have access to a relevant retriever, and therefore marks its prediction as uncertain.

society of LMs, each with access to a different retrieval tool, is presented with a set of factoid questions. As shown in Figure 1, answering correctly requires the agents to know (i) when their parametric knowledge is reliable, (ii) when to express uncertainty to avoid misleading others, and (iii) how to appropriately use their retrieval tools. We hypothesize that training towards this multi-agent setting should encourage better tool use and hedging, which are useful skills even in single-agent scenarios.

To test this hypothesis, we instantiate the game illustrated in Figure 1, and fine-tune a language model on rollouts from the above multi-agent environment using Reinforced Self-Training (ReST; Gulcehre et al., 2023). Specifically, we sample multi-agent interactions from the environment and fine-tune on the rollouts that obtain the highest reward. Crucially, the reward is defined only in terms of *task completion* and *task effort*; we want to achieve high accuracy while avoiding unnecessary tool calls. The actions of individual agents in the rollout are not prescribed or otherwise explicitly rewarded. All agents share parameters (they differ by their prompts and tools), and thus only a single model is trained and evaluated in a *single-agent* setup at test time. Empirically, on BioASQ (Krithara et al., 2023) and PopQA (Mallen et al., 2023), two factoid question answering benchmarks that benefit from corpus-specific retrieval, agents trained using CSP become get better at determining precisely when to search, while learning to expressing uncertainty in a more calibrated manner. This advantage carries over to two unseen benchmarks.

In addition to these experiments, we provide a game-theoretic analysis that sheds light on the conditions under which the optimal strategy for collaborative self-play aligns with the goals of calibrated confidence and tool use, providing concrete guidance for the design of collaborative self-play learning scenarios. To summarize the paper's contributions:

1. We propose a general framework based on collaborative self-play for teaching conversational LM-based agents to be better collaborators when solving tasks.
2. In the context of retrieval-augmented QA, we empirically demonstrate that this framework teaches agents to be better selective tool users and to better express their uncertainty.
3. We provide a game-theoretic analysis that identifies the conditions under which collaborative self-play can induce efficient tool use and calibrated question answering.

## 2  Related work

**Multi-agent LLM systems.** A growing body of work considers systems of multiple language models (for surveys, see Zhuge et al., 2023; Guo et al., 2024), particularly for inference-time deliberation and debate (Du et al., 2023; Chan et al., 2023; Chen et al., 2023; Khan et al., 2024). In contrast, we focus on the impact of multi-agent coordination at the training stage. Along these lines is work in which training examples for reasoning are distilled from multi-agent dialogues (Chen et al., 2024; Subramaniam et al., 2025). We do not work towards specific end-task reasoning but rather we use multi-agent collaboration as a training environment to learn skills that should improve collaboration with humans.

**Social learning.** Our approach can be viewed in the context of *social learning* (Bandura, 1977), which considers the role of socialization in improving predictive performance and

Figure 2: **An example multi-agent rollout**, showing the #ASK, #SEARCH, #HEDGE, and #ANSWER actions, as well as how evidence is incorporated into the prompt. Agent A has no tools, so it asks the other agents for help. Agent B believes its retrieval tool can help, so it performs a search, which returns useful information. Agent C does not have a tool that it believes is applicable. It falls back to parametric knowledge, but marks its answer as a #HEDGE. Agent A receives the two answers, and selects the more confident one, which turns out to be correct.

adaptability, where agents can learn new behaviors by observing and imitating others. For example, Yao et al. (2024) explored how multi-agent collaboration among specialized experts can lead to mutual improvement on image classification tasks, where agents learn from each other to improve their predictions on image classes outside their respective domains of expertise through a group-wide knowledge distillation loss. Similarly, in a LM setting, Mohtashami et al. (2023) use teacher agents to compose instructions or exemplars for student agents, and Wang et al. (2024) use an AI judge to assign reward to the student's social performance. In contrast, we do not designate any agent as a teacher or judge, but rather construct a mechanism in which pro-social behavior is required to get a high reward.

Related ideas are explored in the broader context of multi-agent reinforcement learning (Albrecht et al., 2024), particularly with respect to learning to communicate (Foerster et al., 2016; Sukhbaatar et al., 2016). Here, however, our goal is not to learn an effective multi-agent system, but rather to use the multi-agent context to learn strong single-agent policies — similar to the competitive games underlying GANs and adversarial domain adaptation (Goodfellow et al., 2014; Ganin et al., 2016), but here in a cooperative setting that is furthermore grounded in conversational, natural language. In this respect we deviate from games that are cooperative but non-linguistic (e.g., Bard et al., 2020) or which are linguistic but non-cooperative (e.g., Bakhtin et al., 2022). An additional distinction, which extends to games that are linguistic and at least partially cooperative (e.g., Liao et al., 2024; Xu et al., 2023), is that our goal is not to train language models to succeed in games, but to use games to teach more widely-applicable conversational skills.

**Calibration and confidence.** Language model confidence estimation is an active topic (e.g., Kadavath et al., 2022; Mielke et al., 2022; Yang et al., 2023; Lin et al., 2022; Stengel-Eskin et al., 2024; Quach et al., 2024; Mohri & Hashimoto, 2024), and prior work includes inference-time multi-agent debate (Du et al., 2023; Feng et al., 2024). While some papers claim that LLMs can be prompted to reveal their own uncertainty (Kadavath et al., 2022; Tian et al., 2023, e.g.,), other work raises serious doubts (e.g., Xiong et al., 2023; Kapoor et al., 2024), and in any case, these findings are generally restricted to parametric knowledge rather than the retrieval-augmented models that we consider here. More broadly, the decision to answer or abstain should be driven by two features that are often hard to know in advance: the likelihood that the agent can arrive at the correct answer, and the consequences of abstention vs being incorrect. This view is aligned with Stengel-Eskin et al. (2024), who are also motivated by linguistic pragmatics. But rather than stipulating which action is pragmatically correct in a given context, we create a self-play scenario in which pragmatic reasoning emerges in the solution to the collaborative game. Uncertainty is often decomposed into epistemic (due to lack of knowledge) and aleatoric (due to irreducible randomness; Hüllermeier & Waegeman, 2021). While the subtleties of this distinction are orthogonal to our contribution, we note that our focus is on epistemic uncertainty with regard to (a) the model parameters, and (b) tool outputs, and not, for example, uncertainty about the intent behind the user's query (Min et al., 2020).

**Game-theoretic dialogue.** Prior work offers game-theoretic accounts of various dialogue phenomena, such as implicature (Parikh, 1991; Franke, 2009). We also build on game theory by analyzing the equilibria of a game-theoretic model of collaborative self-play. However, our goal is not to explain features of human conversation but to characterize the conditions under which our proposed mechanism induces normatively-desirable behavior (i.e., calibrated expressions of confidence) from rational agents.

# 3 Social supervision from collaborative self-play

This section describes our framework for obtaining social supervision from collaborative self-play. By *social supervision*, we mean that the training signal emerges from the efficacy of interactions between agents, rather than from direct annotations of reference outputs of individual agents at each step of the rollout. The restriction to *collaborative self-play* is entailed by the use of a single reward for the entire group of agents that all share the same model parameters, rather than distinct rewards and models per agent: all agents are playing together with the same objective to optimize a joint set of parameters.

**Multi-agent orchestration.** For each initial prompt or query $x_1$ we define a rollout as $((a_1, x_1, y_1), (a_2, x_2, y_2), \ldots, (a_T, x_T, y_T))$, with $a_t \in \mathcal{A}$ indicating the active agent at turn $t$ and $x_t, y_t \in \mathcal{V}^*$ indicating its input and output respectively. The initial agent $a_1$ is selected from some initialization policy, and each subsequent agent is then determined by an *orchestrator*, based on the history of the rollout. The rollout continues until either we reach a terminal state, we reach $t = T$, or the rollout exhausts some maximum effort budget, which can be defined in terms of the cost of each action. The terminal state should return an output that can be scored.

The job of the orchestrator is to determine who speaks next: at each step $t < T$, the orchestrator passes control to agent $a_{t+1}$ (which can potentially be the same as $a_t$), and issues a prompt $x_{t+1}$. The new prompt may incorporate tool outputs and communication from other agents. For example, if agent $a_t$ uses a retrieval tool, the orchestrator will add the retrieval results to the prompt $x_{t+1}$ as evidence, while setting $a_{t+1} = a_t$. If agent $a_t$ poses a question, the orchestrator decides on a new $a_{t+1} \neq a_t$ and constructs a prompt $x_{t+1}$ that includes this question. The orchestrator can then remember that the question was posed by $a_t$, and may then return control to this agent after $a_{t+1}$ is finished — or it may pass control to another agent before returning to $a_t$. There are many possible orchestrators, each leading to different patterns of multi-agent interactions: for example, an orchestration policy may allow each agent to broadcast communication to all the others in an egalitarian society, or it may require that all communication flow through a single hub node.

**Actions and arguments.** We now describe an instantiation of collaborative self-play that is designed for training agents to organically learn to use tools and give confidence-calibrated outputs. At each step, each agent's output is required to be an [action] and an [argument], where [action] is limited to the set {#ANSWER, #ASK, #HEDGE} plus actions for each tool available to the agent. This is relatively constrained compared to other implementations of tool use, in which the tool call can appear at any point in the generation (Schick et al., 2023; Mialon et al., 2023); such extensions can be considered in future work. After the [action] and [argument] are sampled from the policy, they are appended to the history, and the orchestrator is called to identify the next agent and its prompt.

When the sampled [action] is a tool, the orchestrator maintains control with the current agent and (potentially) adds evidence to the prompt. We focus on a single type of #SEARCH tool, which obtains a set of near-neighbor retrievals and appends them to the prompt with the preceding tag RETRIEVAL, as shown in Figure 2. A similar approach could be applied to other tools, such as calculators or code interpreters. The remaining actions (i.e., {#ANSWER, #ASK, #HEDGE}) pass control to other agents, as described below.

**Inter-agent communication and orchestration.** Inter-agent communication is enabled by the actions #ASK, #ANSWER, and #HEDGE. If agent $a_t$ issues the #ASK action, then the orchestrator pushes $a_t$ onto a stack $S$ and passes control to a new agent $a_{t+1} \neq a_t$. The argument to the ask action is appended to the prompt $x_{t+1}$, as shown in Figure 2. Agents

---

**Algorithm 1** Action/Answer Reinforced Self-Training

---

**Require:** Initial model $\mathcal{M}_0$, reward threshold $\tau$, number of steps $n_s$, and rollouts per query $n_r$, number of ReST iterations $T$
**Ensure:** Trained model $\mathcal{M}_{\text{ReST}}$
  **for** $t = 1$ **to** $T$ **do**
    **Step 0:** Define an empty training set $\mathcal{D} = \{\}$
    **for** query $q_i$ **do**
      **Step 1a:** Generate rollouts $\{\rho_{i,j}\}_{j=1}^{n_r}$ by applying model $\mathcal{M}_{t-1}$ to $q_i$ in multi-agent orchestration.
      **Step 1b:** Partition the rollouts into action sequences (see § 3), and let $\mathbf{s}_{i,*}$ indicate the action sequence with the maximum mean reward among its compatible rollouts.
      **if** the mean reward of $\mathbf{s}_{i,*} > \tau$ **then**
        **Step 1c:** Among rollouts compatible with $\mathbf{s}_{i,*}$, select the one with the highest reward, $\rho_{i,*} = \arg\max_j\{r_{i,j} : \rho_{i,j} \vdash \mathbf{s}_{i,*}\}$.
        **Step 1d:** Convert $\rho_{i,*}$ into turn-level training examples and add them to $\mathcal{D}$.
      **end if**
    **end for**
    **Step 2:** Train $\mathcal{M}_t$ on a training split of $\mathcal{D}$ for $n_s$ steps.
    **Step 3:** Select the checkpoint with the smallest held-out log-likelihood on a dev split of $\mathcal{D}$.
  **end for**
  **return** $\mathcal{M}_T$

---

relinquish control by issuing the actions #ANSWER and #HEDGE. The orchestrator can then return control to the agent at the top of the ask stack, or, in a "broadcast" setting, can pass control to another agent. When control returns to $a_t$, its prompt is updated with the string arguments provided by the answering agent(s), with the preceding tag FRIEND_ANSWER or FRIEND_HEDGE (see Figure 2). If the ask stack is empty when a control-relinquishing action is issued, the rollout has entered a terminal state.

We implement a broadcast version of the #ASK action. The action causes the ask stack to be updated as $S_t \leftarrow a_t \circ S_{t-1}$. Control then passes to each agent reachable from $S_t$. Conversely, if $a_t$ issues a control-relinquishing action with ask stack $S_{t-1} = a \circ S'$, then control passes to agent $a$ with ask stack $S_t = S'$. We reach a terminal state when the initial agent $a_1$ relinquishes control and the stack $S_T$ is empty. Reward is then computed as effort-penalized task performance, i.e., we would like to return a correct answer, while performing as few tool calls as possible. Specifically, $r(h_T) = Score(y_T, y^*) - \delta \cdot Effort(h_T)$, where *Score* is a metric like token-level $F_1$, effort is the number of search calls in the rollout, and $\delta$ is a hyper-parameter.

**Training.** Given a society, an orchestrator, and a reward function, we hope to learn language model policies that yield high expected reward. We apply Reinforced Self-Training (ReST), an extension of supervised learning that is commonly used in post-training (e.g., Gulcehre et al., 2023; Zhang et al., 2024). The idea is to generate rollouts, score them, train on the good ones, and then iterate with the newly trained policy. However, the application of ReST to collaborative self-play requires adaptation to avoid fine-tuning on lucky guesses that do not lead to generally effective policies.

Specifically, we will focus first on identifying sequences of *actions* that reliably lead to good rewards, and then select the best *rollout* compatible with such action sequences (see Algorithm 1). Recall that a rollout is defined as $\rho = \{(a_t, x_t, y_t)\}_{t=1}^T$, with $y_t = [z_t, c_t]$ now composed of an action $z_t \in \mathcal{Z}$ and an optional argument $c_t$. An action sequence $\mathbf{s} = (s_1, s_2, \ldots)$ is a sequence of actions, $s_t \in \mathcal{Z}$ (such as #ANSWER or #SEARCH). If for a given rollout $\rho$ we have $z_t = s_t$ for all $t$, then we write $\rho \vdash \mathbf{s}$ to indicate that $\rho$ is *compatible* with $s$. Now a set of rollouts, $\{\rho_1, \rho_2, \ldots\}$ can be coarsened into a set of action sequences $\{\mathbf{s}_1, \mathbf{s}_2, \ldots\}$. Each action sequence is scored by the average reward of its compatible rollouts, and the best action sequence $\mathbf{s}^*$ is selected. If this average reward is above some threshold, we then choose the best individual rollout $\phi^* \vdash \mathbf{s}^*$, and train on each step in this rollout. In ReST, training is done iteratively, and we run it for $T$ epochs, where in each epoch rollouts are sampled from the mostly recently trained model.

Importantly, while we have a multi-agent environment, all agents share the same parameters and only differ in their tool access and input prompt, and thus we train a single model. This is necessary because our goal is to use the multi-agent environment to elicit training data that will improve the model, but then have a single agent at test time.

|           | $Z_2 = S$                                         | $Z_2 = G$                        |
| --------- | ------------------------------------------------- | -------------------------------- |
| $Z_1 = S$ | $(\overline{\alpha} - \delta, \overline{\alpha} - \delta)$ | $(\alpha_1 - \delta, \alpha_1)$ |
| $Z_1 = G$ | $(\alpha_2, \alpha_2 - \delta)$                   | $(\beta, \beta)$                 |

Table 1: Payoff matrix for the game described in § 4, where each cell shows the reward for the two players, $(r_1, r_2)$.

# 4 Analysis: An information-provision game

Our core intuition is that the group can achieve a high reward only by learning to calibrate its use of the #ANSWER and #HEDGE actions. To clarify the necessary conditions for this to work, we analyze the pure strategy equilibria of a simplified two-player game, where an asker agent issues a question to two players with access to different retrieval tools, and each player has the choice between actions $S$ (search-then-answer) and $G$ (guess-then-hedge). Agent $i$ generates the correct answer with probability $\alpha_i$ when playing action $Z_i = S$, and $\beta$ when playing action $Z_i = G$. When one player plays $Z_i = S$ and the other player plays $Z_j = G$, the asker chooses the more confident answer (from player $i$); otherwise the asker picks one of the two answers at random. Both players receive a reward of 1 if the question is answered correctly and zero otherwise; the search tool incurs a cost of $\delta > 0$. Table 1 shows the *expected* rewards under the joint distribution associated with the correctness probabilities $\alpha_1, \alpha_2, \beta$ and the random choice made by the asker, with $\overline{\alpha} = \frac{1}{2}(\alpha_1 + \alpha_2)$.[1] We analyze this game to find the parametrizations under which there is a single equilibrium corresponding to our normative expectation of the agents' behavior under full information: i.e., the use of the $S$ action should indicate that the agent is especially likely to provide a correct answer.

**Theorem 4.1.** *Assume without loss of generality that $\alpha_1 > \overline{\alpha} > \alpha_2$, $\{\alpha_1, \alpha_2\} \neq \beta + \delta$, and $\alpha_1 \neq \alpha_2 + 2\delta$. Then the game in Table 1 has a unique pure strategy equilibrium if and only if at least one of two conditions is met: $\alpha_1 > \alpha_2 + 2\delta$ or $\alpha_2 < \beta + \delta$. Specifically, the unique equilibrium is $(S, G)$ if $\alpha_1 > \beta + \delta$ and $(G, G)$ otherwise. If neither condition is met, then $(S, G)$ and $(G, S)$ are both pure strategy equilibria.*

The proof is in Appendix C, which also includes a generalization to $n$-player games. The strategy space is illustrated in Figure 6 in Appendix D. The analysis sheds light on what the information-provision game can teach, and provides guidance on how to design the game to achieve the desired results. We are most interested in settings where $(S, G)$ is the unique equilibrium, so that the players' actions are informative of their relative accuracies. This will occur if the players have search tools that are very effective on some questions, $\alpha_i(x) \gg \beta(x)$, with $\alpha_i(x)$ and $\beta(x)$ indicating the probability of answering correctly on a given question $x$,[2] and when the tools are *complementary* in the sense that if $\alpha_1(x) \gg \beta(x)$ then $\alpha_1(x) \gg \alpha_2(x)$, so that it is clear which tool to use. If $\alpha_2(x) \approx \beta(x)$ then the agents will learn to search precisely when it is better than guessing. In the next section we describe a game that approximately meets these criteria.

# 5 Experimental setup

We evaluate whether collaborative self-play can teach agents when (a) they can answer confidently from parametric knowledge; (b) they have a tool that is likely to yield helpful evidence; or (c) they cannot answer confidently and must therefore hedge.

**Evaluation details.** Our evaluation focuses on a practical special case of collaborative self-play (defined in § 3), using the Gemma2-9b base model (Gemma Team et al., 2024).

- **Helper agents**: WIKI-BM25, which can search Wikipedia using BM-25 (Robertson & Zaragoza, 2009); PUBMED-GECKO, which can search PubMed abstracts using dense embeddings from Gecko (Lee et al., 2024). When using the #SEARCH tool, the agents must pass along the query verbatim. Helper agents cannot issue an #ASK action.

---

[1]This game is similar to the classical example of the provision of a public good (see, e.g., Olson Jr, 1971; Osborne, 2004, Chapter 2), where the good is the information provided by the search tool.

[2]These conditional probability models are assumed to be learned by minimizing regret.

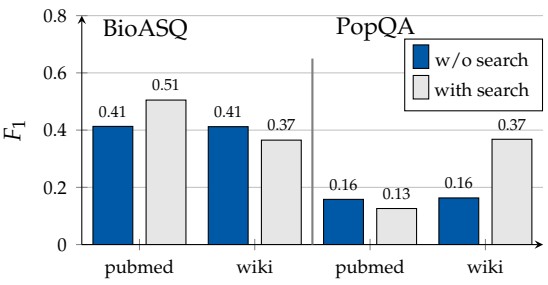

Figure 3: **Mean $F_1$ of the prompted model for each helper agent**. The tool-augmented agents have complementary knowledge, satisfying the conditions for collaborative self-play calibration described in § 4.

- **Communication**: An asker agent that has no tool access passes a query to the helper agents by issuing an #ASK action; their responses enter the evidence section of the asker's prompt as either #FRIEND_ANSWER or #FRIEND_HEDGE. For #ASK and #SEARCH the argument is copied from the last turn, while for #ANSWER and #HEDGE it is generated by the model.

- **Data**: Short-answer questions from BioASQ (Krithara et al., 2023) and PopQA (Mallen et al., 2023). BioASQ is a manually-curated corpus of biomedical questions along with gold answers. PopQA is an open-domain question answering dataset with questions on entities of various levels of popularity from Wikipedia. We choose these two benchmarks guided by the analysis from § 4, as they have long-tail questions that are less likely to be in the model's parameteric knowledge, and thus the advantage of tool use should be considerable for some questions.[3]

- **Out-of-distribution evaluation**: We evaluate on the dev sets of two additional benchmarks: Natural Questions (NQ; Kwiatkowski et al., 2019) and EntityQuestions (EntQ; Sciavolino et al., 2021). Both contain questions answerable from Wikipedia. However, NQ questions focus on common entities, so are likely answerable from parametric knowledge, while EntQ focuses more on tail knowledge that is likely to require retrieval.

- **Reward**: Effort-penalized $F_1$ (comparing the set of tokens in the gold and predicted answers), with a penalty of $\delta$ for each tool use call. The penalty is set to a low value to serve as a tie-breaker between rollouts with the same $F_1$.

An example rollout in this setting is shown in Figure 2; the full prompts are in Appendix A.

Why does this implementation of collaborative self-play teach the capabilities enumerated above? As argued in § 4, this setup will incentivize calibration of the #ANSWER and #HEDGE actions if the agents have complementary knowledge, because using the correct action is necessary to enable the asker to choose effectively between conflicting answers provided by the helpers. As shown in Figure 3, the agents do indeed have complementary knowledge, with significant gaps in $F_1$ across the two datasets. Furthermore, the prompted Gemma2-9b model attends to the #ANSWER/#HEDGE distinction: in 80% of rollouts where it receives an #ANSWER and a #HEDGE from its helper agents, it passes along the more confident answer. There is also an incentive for efficient tool use: if the agents can obtain an accurate answer parametrically, they will achieve the highest reward by doing so. They should make tool calls only if they are likely to yield good evidence, or they will incur an effort penalty without improving their likelihood of correctness.

**Models and baselines.** We compare the following supervision strategies:

- **In-context learning (ICL)**. We use the base gemma2-9B model, with one-shot-per-action prompt shown in Appendix A.

- **Collaborative self-play ReST (CSP)**. We generate rollouts from the three-agent society and train as in Algorithm 1, with $\tau = 0.1, n_s = 2000, n_r = 32$.[4] Hedging and search are learned only from inter-agent communication.

- **Deanonymized CSP ReST (CSP-DeAnon)**. Similar to CSP, but the asker agent knows the identities of the helpers. As an example, in the last turn of Figure 2 we replace FRIEND_ANSWER with FRIEND_ANSWER (wiki), and FRIEND_HEDGE with FRIEND_HEDGE

---

[3]In PopQA, we filter out questions with higher than median annotated popularity.
[4]Pilot experiments with $\tau = 0.5, n_s \in \{1000, 5000\}, n_r \in \{24, 100\}$ yielded broadly similar results.

| setting | agent | task-level $F_1$ ↑ | | | | search rate ↓ | | | |
|---|---|---|---|---|---|---|---|---|---|
| | | bioasq | popqa | nq | entq | bioasq | popqa | nq | entq |
| ICL | PUBMED | 0.577 | 0.112 | 0.267 | 0.212 | 0.984 | 0.977 | 0.967 | 0.980 |
| | WIKI | 0.423 | 0.338 | 0.407 | 0.521 | 0.972 | 0.995 | 0.978 | 0.994 |
| CSP | PUBMED | 0.568 | 0.235 | 0.377 | 0.329 | 0.228 | 0.191 | 0.104 | 0.126 |
| | WIKI | 0.538 | 0.321 | 0.404 | 0.434 | 0.182 | 0.480 | 0.205 | 0.362 |
| CSP-DeAnon | PUBMED | 0.576 | 0.234 | 0.386 | 0.349 | 0.251 | 0.091 | 0.059 | 0.052 |
| | WIKI | 0.537 | 0.325 | 0.391 | 0.449 | 0.140 | 0.499 | 0.190 | 0.344 |
| Act. Sup. | PUBMED | 0.568 | 0.243 | 0.375 | 0.332 | 0.180 | 0.039 | 0.063 | 0.060 |
| | WIKI | 0.542 | 0.293 | 0.395 | 0.373 | 0.136 | 0.188 | 0.092 | 0.131 |
| Oracle | PUBMED | 0.690 | 0.256 | 0.428 | 0.384 | 0.294 | 0.048 | 0.079 | 0.075 |
| | WIKI | 0.606 | 0.427 | 0.537 | 0.619 | 0.175 | 0.248 | 0.209 | 0.379 |

Table 2: **Task performance and effort.** On in-distribution data (bioasq and popqa), collaborative self-play (CSP) achieves higher or similar task-level $F_1$ compared to in-context learning (ICL), despite using 2-5x fewer search calls. The action-supervised method is able to further reduce search rates by training directly on a calibration-based objective, but does not improve the task-level $F_1$ significantly. On EntQ, an OOD dataset, Action Supervision searches too little, reducing task performance.

    (pubmed). This means that the asker can learn which helper is likely to give a correct answer for a particular question (ignoring any of the helper's confidence markers). This reduces the incentive for the helper agents to learn to hedge in CSP-DeAnon, which we predict will reduce the calibration of $P(\texttt{\#ANSWER})$.

- **Action supervision**. We generate rollouts from the prompted base model, with no inter-agent communication. We then construct a silver label for the optimal action according to a calibration-based objective. Specifically, for each question, we take the max-reward rollout that ended with #ANSWER. If the $F_1 > 0.5$, we train on the steps of this rollout; otherwise, we take the highest-reward rollout that ended with #HEDGE, and train on its actions if its $F_1 < 0.5$, excluding the answer. Action supervision directly maps #ANSWER to high-$F_1$ rollouts and #HEDGE to low-$F_1$ rollouts, while CSP-ReST relies on task completion alone.

The evaluation focuses on a *single-agent deployment scenario* with #ASK disabled, testing the ability of multi-agent supervision to transfer to single-agent policies. Thus, we run the model in a single-agent setup, where the agent chooses between #ANSWER, #HEDGE, and #SEARCH. The goal is to use #ANSWER when it is possible to answer from parametric knowledge, #SEARCH when it is likely to be helpful, and #HEDGE when unsure.

Action Supervision is more similar to standard calibration strategies (e.g., Lin et al., 2022; Zhang et al., 2023), where we teach the model to be confident when it is correct and unsure when is wrong. Its training is closely aligned with the evaluation procedure (see § 6), heuristically splitting #ANSWER vs #HEDGE on $F_1 > 0.5$. Conversely, CSP does not prescribe any specific actions, relying on the emergent training signal from task completion alone. Thus Action Supervision can be regarded as an upper bound on the calibration performance attainable by CSP on these tasks.

**Inference cost to generate training data.** Generating the training data required 64,000 rollouts per round of ReST training: 1000 queries per dataset, 32 rollouts per query, two datasets), with three to five inference calls per rollout. Empirically, each ReST epoch produced $< 2M$ whitespace-delimited output tokens, from $< 180M$ whitespace-delimited input tokens. Over three epochs and with an approximate multiplier of 1.3 subword tokens per whitespace token, we estimate the total cost of replicating the training data generation at less than $150 US, based on third-party prices for gemma2-9b inference.[5]

---

[5] https://groq.com/pricing/, showing US$0.20 per million tokens on August 8, 2025.

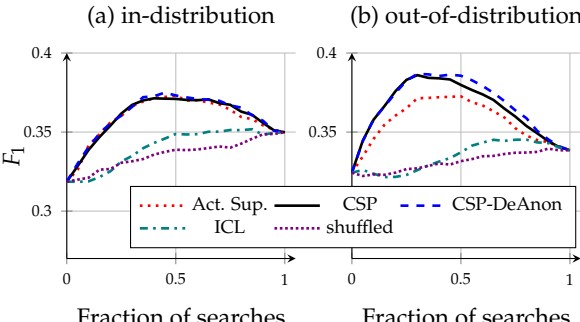

(a) in-distribution     (b) out-of-distribution

Figure 4: **Calibration of** $P(\texttt{\#SEARCH})$. For each method, held-out queries are sorted by $P(\texttt{\#SEARCH})$; in 'shuffled' they are randomly shuffled. For the top $x\%$ of queries, retrieval is used; for the remainder, the model must respond parametrically. Well-calibrated tool users can achieve significant boosts in $F_1$ from even rare use of #SEARCH.

## 6 Results

We evaluate the learned agents on three dimensions: (1) whether they can reliably obtain accurate answers at minimal cost; (2) whether their use of the search tool is calibrated to its helpfulness; (3) whether their use of #ANSWER vs #HEDGE is calibrated to correctness. Here we report results after three ReST epochs; for per-epoch results see Appendix E.

**Task performance.** To measure task performance, we run each helper agent (PubMED-Gecko and Wiki-BM25) on held-out questions from each dataset, and report token-level $F_1$. Results are shown in Table 2. The most prominent distinction between the methods is that in-context learning (ICL) nearly always searches, while the three supervised methods search less than half of the time in most settings — usually much less. As a result, ICL yields poor performance when applied to an agent whose tool is not suited for the dataset (e.g., Wiki/BioASQ, PubMed/PopQA), with $F_1$ gaps of more than 10 points from the other techniques. Among the supervised methods, $F_1$ results are broadly similar, with Action Supervision searching at the lowest rate due to its calibration-based objective. However, this low search rate hurts Action Supervision on the EntQ dataset: on Wiki/EntQ it trails CSP by 6.1 $F_1$. Conversely, ICL performs especially well on this dataset, because of its high search rate ($> 99\%$). The deanonymization ablation has relatively little impact on task performance, except that CSP-DeAnon is able to reduce the search rate in mismatched settings.

The table also shows the performance of a *test-time* oracle, which uses the retrieval tool on exactly those instances where it improves $F_1$. This shows that there remains 10 points of $F_1$ headroom for in-domain agents at slightly higher search rates.

**Search calibration.** A key capability that we hope to teach from collaborative self-play is when to retrieve and when to rely on parametric knowledge. To measure this, we sort questions by $P(\texttt{\#SEARCH})$ according to each model. At each quantile threshold $\tau$, we apply retrieval to all questions where $P(\texttt{\#SEARCH}) > \tau$, and answer the remaining questions parametrically. To focus on evaluation of $P(\texttt{\#SEARCH})$ specifically, the answers themselves are drawn from the base model rather than the finetuned model. This isolates the impact of search calibration from other aspects of the question-answering capability.

In-distribution results (BioASQ and PopQA) are shown in Figure 4 (a). On the far left, search is applied only to a very small number of queries; on the far right it is applied to every query. Agents with strong search calibration will display a rapid increase in mean $F_1$ as the fraction of searches increases from zero, and will show a decrease in $F_1$ as we approach the far right side of figure, because irrelevant retrievals makes QA less accurate (see Figure 3). As shown in the figure, all trained models are better calibrated than ICL, which barely outperforms a baseline that randomly shuffles the questions.

Out-of-distribution results (NQ and EntQ) are shown in Figure 4 (b). Here, we see that Action Supervision is less robust than Collaborative Self-Play (CSP), with worse generalization to the OOD setup. CSP-DeAnon slightly outperforms CSP at relatively high search rates, but will be shown to have much lower $P(\texttt{\#ANSWER})$ calibration.

**Answer calibration.** Finally, we explore calibration with respect to parametric knowledge. To do this, we compare the probabilities of answering and guessing without search, sort-

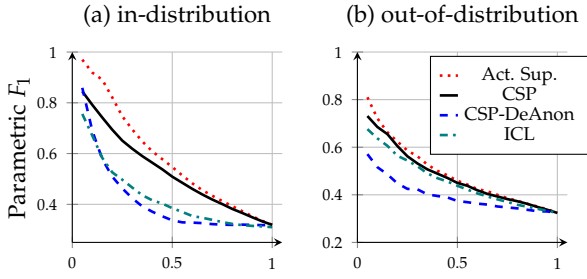

(a) in-distribution  (b) out-of-distribution

Frac. questions answered  Frac. questions answered

Figure 5: **Calibration of P(#ANSWER) vs P(#HEDGE).** For each method, held-out queries are sorted by $P(\texttt{\#ANSWER})/(P(\texttt{\#ANSWER}) + P(\texttt{\#HEDGE}))$. For the top $x\%$ of queries, we compute the mean $F_1$ of the prompted parametric model. Deanonymization significantly reduces calibration.

| Setting | In-dist. | OOD |
|---|---|---|
| CSP | 0.5363 | 0.3663 |
| CSP-DeAnon | 0.0696 | 0.1677 |
| Action Supervision | 0.6637 | 0.3991 |
| ICL | 0.2314 | 0.3179 |

Table 3: Spearman correlations between P(#ANSWER) and parametric $F_1$, quantifying the relationship shown in Figure 5. Anonymization is the key feature of the mechanism by which calibration is learned from collaborative self-play.

ing by $P(\texttt{\#ANSWER})/(P(\texttt{\#ANSWER}) + P(\texttt{\#HEDGE}))$. Note that this form of calibration may not emerge from the collaborative self-play game, because the agents have identical parametric knowledge, and can only obtain an advantage by using their corpus-specific retrieval tools. On the other hand, action supervision is expected to teach this capability, because the model is directly trained to use #ANSWER when it can produce a high $F_1$.

As shown in Figure 5, Action Supervision does indeed yield the best calibration of $P(\texttt{\#ANSWER})$. CSP is slightly behind, but substantially outperforms both ICL and CSP-DeAnon. The gap between CSP and CSP-DeAnon validates the intuition and theory presented above: a small change in the prompt (revealing the identity of the helper) leads to a large gap in calibration, because the change breaks the mechanism linking calibration to performance on the collaborative self-play task. On the OOD benchmarks, the gap between Action Supervision, CSP, and ICL is small, with CSP-DeAnon trailing significantly.

**Learning dynamics.** Learning dynamics for the three epochs of reinforced self-training (ReST) are shown in Appendix E. To briefly summarize, the search rate decreases consistently throughout training, as the policy works to maintain task performance while incurring lower cost, while task performance remains nearly constant after the first epoch.

## 7 Discussion

This work shows that calibration and efficient tool use can emerge from relatively weak supervision through a multi-agent collaborative game. Given the right incentive structure, agents learn when their corpus-specific retrieval tool can provide useful additional information, and then respond with a binary confidence signal. This suggests that mechanism design and multi-agent interaction can offer an alternative approach for post-training towards language models with stronger interactional capabilities.

**Limitations.** This paper considers only retrieval tools, factoid QA tasks, and a single language model. We hope to relax all of these limitations, for example by considering alternative tools (e.g., calculators, code interpreters) and more complex inter-agent interactions (e.g., question decomposition). We are particularly curious to apply CSP to tasks beyond factoid QA, particularly in tasks in which ground truth validators are unavailable. Longer-term, we would like to generalize to the approach other grounding strategies, such as requesting and issuing clarifications.

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

## A Prompts

We attach the full prompts below for the Wiki-BM25 agent (before and after #SEARCH was applied) and for the no-tools agent after receiving answers from the helper agents.

```
You are a helpful agent whose job is to answer a question using verified information.

To answer to the question, you can execute the following actions:

**SEARCH**: search a corpus of a set of passages from Wikipedia pertaining to facts about famous people, locations,
↪  and historical events to find relevant information.
**HEDGE**: guess an answer if you think you might know but are not sure and have no way to find out more.
**ANSWER**: directly answer the question when you are confident that you have the correct answer.

Each of these actions takes a string argument, which are:

**SEARCH**: the argument is the search query.
**HEDGE**: the argument is the guess for the question.
**ANSWER**: the argument is the answer to the question.

Here are some examples.

Example: Using **ANSWER** directly.
QUESTION: what is the name of a figure with three sides?
ACTION: **ANSWER**: a triangle

Example: Using **SEARCH** to search a corpus for evidence.
QUESTION: who is the starting center for the Denver Nuggets?
ACTION: **SEARCH**: who is the starting center for the Denver Nuggets?

Example: Using **ANSWER** after getting evidence from searching a corpus.
QUESTION: who is the starting center for the Denver Nuggets?
RETRIEVAL: Nikola Jokic is the starting center for the Denver Nuggets.
RETRIEVAL: Joel Embid is the starting center for the Philadelphia 76ers.
RETRIEVAL: Topeka is the geographical center of the United States.
QUESTION: who is the starting center for the Denver Nuggets?
ACTION: **ANSWER**: Nikola Jokic

Example: Using **HEDGE** to guess an answer when unsure.
QUESTION: what is the name of a figure with eleven sides?
ACTION: **HEDGE**: a endecagon

You should use **SEARCH** if you do not know the answer and think your corpus is likely to contain useful
↪  information.
You should use **HEDGE** if you think you might know the answer but are not fully confident, and cannot get more
↪  useful information. Do not give hedged answers or say 'I do not know', just make your best guess and mark your
↪  confidence by using the **HEDGE** prefix.
You should use **ANSWER** only if you are very confident that you will be correct, based on the evidence you have
↪  obtained.

Your answer should be short. For example, if the question is "What is the capital of France?", please answer
↪  "Paris", and not "Paris is the capital of France". If you are asked a yes/no question, you may only answer
↪  "yes" or "no". Do not give hedged answers like "maybe". Instead, you can use **HEDGE** to indicate low
↪  confidence.

Always respond in a single line with the format "ACTION: **<the action>**: <the argument>"

QUESTION: Who is the author of The Girl?
```

You are a helpful agent whose job is to answer a question using verified information.

To answer to the question, you can execute the following actions:

**SEARCH**: search a corpus of a set of passages from Wikipedia pertaining to facts about famous people, locations,
↪  and historical events to find relevant information.
**HEDGE**: guess an answer if you think you might know but are not sure and have no way to find out more.
**ANSWER**: directly answer the question when you are confident that you have the correct answer.

Each of these actions takes a string argument, which are:

**SEARCH**: the argument is the search query.
**HEDGE**: the argument is the guess for the question.
**ANSWER**: the argument is the answer to the question.

Here are some examples.

Example: Using **ANSWER** directly.
QUESTION: what is the name of a figure with three sides?
ACTION: **ANSWER**: a triangle

Example: Using **SEARCH** to search a corpus for evidence.
QUESTION: who is the starting center for the Denver Nuggets?
ACTION: **SEARCH**: who is the starting center for the Denver Nuggets?

Example: Using **ANSWER** after getting evidence from searching a corpus.
QUESTION: who is the starting center for the Denver Nuggets?
RETRIEVAL: Nikola Jokic is the starting center for the Denver Nuggets.
RETRIEVAL: Joel Embid is the starting center for the Philadelphia 76ers.
RETRIEVAL: Topeka is the geographical center of the United States.
QUESTION: who is the starting center for the Denver Nuggets?
ACTION: **ANSWER**: Nikola Jokic

Example: Using **HEDGE** to guess an answer when unsure.
QUESTION: what is the name of a figure with eleven sides?
ACTION: **HEDGE**: a endecagon

You should use **SEARCH** if you do not know the answer and think your corpus is likely to contain useful
↪  information.
You should use **HEDGE** if you think you might know the answer but are not fully confident, and cannot get more
↪  useful information. Do not give hedged answers or say 'I do not know', just make your best guess and mark your
↪  confidence by using the **HEDGE** prefix.
You should use **ANSWER** only if you are very confident that you will be correct, based on the evidence you have
↪  obtained.

Your answer should be short. For example, if the question is "What is the capital of France?", please answer
↪  "Paris", and not "Paris is the capital of France". If you are asked a yes/no question, you may only answer
↪  "yes" or "no". Do not give hedged answers like "maybe". Instead, you can use **HEDGE** to indicate low
↪  confidence.

Always respond in a single line with the format "ACTION: **<the action>**: <the argument>"

QUESTION: Who is the author of The Girl?
RETRIEVAL: title: Kulpreet Yadav passage: he was awarded the Director General's Commendation for professionalism
↪  and dedication to the nation. He retired voluntarily in the rank of Commandant with the Indian Coast Guard in
↪  2014. Kulpreet lives in Delhi with his wife Seema and daughters Leah and Jeanie. Kulpreet Yadav Kulpreet Yadav
↪  is an Indian writer in the fiction-Thriller genre. He is the author of two novels: ""The Girl Who Loved a
↪  Pirate"" and ""The Girl Who Loved a Spy"". ""The Girl Who Loved a Pirate"" is India's first thriller based on
↪  marine piracy & hijacking. Kulpreet was born in Chennai and completed graduation in Science
RETRIEVAL: title: Kulpreet Yadav passage: Kulpreet Yadav Kulpreet Yadav is an Indian writer in the
↪  fiction-Thriller genre. He is the author of two novels: ""The Girl Who Loved a Pirate"" and ""The Girl Who
↪  Loved a Spy"". ""The Girl Who Loved a Pirate"" is India's first thriller based on marine piracy & hijacking.
↪  Kulpreet was born in Chennai and completed graduation in Science from Nowrosjee Wadia College, Pune. He
↪  completed his post-graduation in Journalism and Mass Communication from Amity University, Noida in 2004 and
↪  Management courses from IIM, Indore and IIM, Lucknow. He joined the Naval Officer's Academy and served for two
↪  decades. In 2007
RETRIEVAL: title: The Simple Girl passage: The film's sets were designed by the art directors Emil Hasler and Paul
↪  Markwitz. The film premiered on 23 August 1957 at the Thalia in Wiesbaden. Caterina Bastiani, a talented young
↪  actress, is offered the leading role in a musical. This is her big break but the author of the novel on which
↪  the musical is based is less than pleased about this adaption | and he does not think much of Caterina.
↪  Caterina meets a girl by accident who has applied to work for the author as a maid. She takes the girl's place
↪  in order to prove her
RETRIEVAL: title: Peter Leonard (author) passage: is the author of: - Quiver - Trust Me - All He Saw Was The Girl -
↪  Voices of the Dead - Back from the Dead (sequel to Voices of the Dead) - Eyes Closed Tight - Unknown Remains
↪  Peter Leonard (author) Peter Leonard, the son of Elmore Leonard, is an American author of crime novels. In
↪  1980, Peter was the founding partner of the advertising agency Leonard Mayer & Tocco. For nearly thirty years
↪  LM&T created award-winning advertising for Volkswagen of America, Audi of America, Hiram Walker, and Pennzoil.
↪  He wrote his first novel, ""Quiver"", in 2007; he has
RETRIEVAL: title: Zoe Strimpel passage: became normalised. Strimpel is the author of ""What the Hell is He
↪  Thinking?: All the Questions You've Ever Asked About Men Answered"", which was published in July 2010. It is
↪  aimed at providing an insight into men's thinking, researched by Strimpel interviewing men. Her second book,
↪  ""The Man Diet: One Woman's Quest to End Bad Romance"" was published on 22 December 2011. Both books received
↪  positive reviews from critics and press coverage. Strimpel originally wrote for ""The Times"" as a freelancer.
↪  From 2006, she was the author of the ""Girl about town"" column in ""The London Paper"", a now-defunct free
QUESTION: Who is the author of The Girl?

```
You are a helpful agent whose job is to answer a question using verified information.

To answer to the question, you can execute the following actions:

**ANSWER**: directly answer the question when you are confident that you have the correct answer.
**HEDGE**: guess an answer if you think you might know but are not sure and have no way to find out more.
**ASK**: ask your friends if they might know how to answer the question.

Each of these actions takes a string argument, which are:

**ANSWER**: the argument is the answer to the question.
**HEDGE**: the argument is the guess for the question.
**ASK**: the argument is the question you ask your friend.

Here are some examples.

Example: Using **ANSWER** directly.
QUESTION: what is the name of a figure with three sides?
ACTION: **ANSWER**: a triangle

Example: Using **ASK** to ask a friend.
QUESTION: who is the starting center for the Denver Nuggets?
ACTION: **ASK**: who is the starting center for the Denver Nuggets?

Example: Using **ANSWER** after getting evidence from asking a friend.
QUESTION: who is the starting center for the Denver Nuggets?
FRIEND'S ANSWER: Nikola Jokic
QUESTION: who is the starting center for the Denver Nuggets?
ACTION: **ANSWER**: Nikola Jokic

Example: Using **HEDGE** to guess an answer when unsure.
QUESTION: what is the name of a figure with eleven sides?
ACTION: **HEDGE**: a endecagon

You should use **ANSWER** only if you are very confident that you will be correct, based on the evidence you have
↪  obtained.
You should use **HEDGE** if you think you might know the answer but are not fully confident, and cannot get more
↪  useful information. Do not give hedged answers or say 'I do not know', just make your best guess and mark your
↪  confidence by using the **HEDGE** prefix.
You should use **ASK** if you think your friend might have the information you lack.

Your answer should be short. For example, if the question is "What is the capital of France?", please answer
↪  "Paris", and not "Paris is the capital of France". If you are asked a yes/no question, you may only answer
↪  "yes" or "no". Do not give hedged answers like "maybe". Instead, you can use **HEDGE** to indicate low
↪  confidence.

Always respond in a single line with the format "ACTION: **<the action>**: <the argument>"

QUESTION: In which fields of DNA sequencing are Bloom filters applied?
FRIEND'S ANSWER: error analysis, storage optimization
FRIEND'S HEDGE: pattern matching, lossless compression, host species sequence screening, k-mer counting
QUESTION: In which fields of DNA sequencing are Bloom filters applied?
```

## B  Terminology

To clarify the discussion, we offer the following definitions. Although the general framework does not require agents to be tool users, our application of the framework will require tools, and therefore we include the relevant definitions here.

**Definition B.1.** *A **tool** is defined by a keyword and a function $f : \mathcal{V}^* \to \{\mathcal{V}^*\}^*$ from an argument string to a list of output strings.*

For example, one tool might take the keyword #SEARCH and a query, and return a list of snippets of retrieved Wikipedia pages; another tool might take the keyword #CALCULATE and an arithmetic expression, and return an arithmetic result (Parisi et al., 2022; Gao et al., 2023).

**Definition B.2.** *An **agent** $a \in \mathcal{A}$ is defined by a unique identifier and by a (possibly empty) list of accessible tools.*

While the mapping between agents and tools is potentially many-to-many, we will focus on cases in which each tool is assigned to exactly one agent (e.g., there is only one agent who is able to #SEARCH Wikipedia pages).

**Definition B.3.** *A **society** $S \in \mathcal{S}$ is defined by a set of directed pairs of agents, $S = \{(a_{i,1}, a_{i,2})_{i=1}^N\}$. If $(a_i, a_j) \in S$ then $a_i$ can pass control and information to $a_j$.*

For example, a society may be fully or sparsely connected, symmetric or asymmetric. The way that control and information between agents is passed along in the society is then handled by a dedicated orchestrator.

**Definition B.4.** *An **orchestrator** $O : \mathcal{S} \times \mathcal{H} \to \mathcal{A} \times \mathcal{V}^*$ is a function from a society $S \in \mathcal{S}$ and a history of prompts and outputs $\{(x_i, y_i)_{i=1}^t\} \in \mathcal{H}$ to an agent $a_{t+1} \in \mathcal{A}$ and a prompt $x_{t+1} \in \mathcal{V}^*$.*

The orchestrator may be deterministic or random, and may be arbitrarily complex. In our setting, the agent's policies are learned, but the orchestrator is not. Even for a fixed orchestrator, the patterns of interactions between agents can change dramatically over the course of training.

**Definition B.5.** *A **reward** $r : \mathcal{H}_T \to \mathbb{R}$ assigns a scalar score to a history of length T of prompts and outputs that has reached a terminal state.*

Intuitively, the reward is a measure of how effective the society was at functioning in the current history (e.g., in response to an initial prompt or query). For example, a reward might score the accuracy of the final output $y_T$ against some reference; it might include a penalty for the length of the rollout or the number of tool calls; it could be a learned reward model trained from preference annotations; or it might be the output of a prompted auto-evaluator.

## C  Additional theory

**Theorem C.1.** *Assume without loss of generality that $\alpha_1 > \overline{\alpha} > \alpha_2$ and that $\{\alpha_1, \alpha_2\} \neq \beta + \delta$ and $\alpha_1 \neq \alpha_2 + 2\delta$. Then the game in Table 1 has a unique pure strategy equilibrium if and only if at least one of two conditions is met: $\alpha_1 > \alpha_2 + 2\delta$ or $\alpha_2 < \beta + \delta$. Specifically, the unique equilibrium is $(S, G)$ if $\alpha_1 > \beta + \delta$ and $(G, G)$ otherwise. If neither condition is met, then $(S, G)$ and $(G, S)$ are both pure strategy equilibria.*

*Proof.* In all cases, $r_2(S, G) > r_2(S, S)$ because $\alpha_1 > \overline{\alpha} - \delta$ by construction. If $\alpha_1 > \alpha_2 + 2\delta$ then $r_1(S, S) > r_1(G, S)$. The unique equilibrium is then $(S, G)$ if $\alpha_1 > \beta + \delta$, and $(G, G)$ if $\alpha_1 < \beta + \delta$, because $\alpha_1 < \beta + \delta \Rightarrow \alpha_2 < \beta + \delta \Rightarrow r_2(G, S) < r_2(G, G)$.

If $\alpha_1 < \alpha_2 + 2\delta$ then $r_1(G, S) > r_1(S, S)$. Then $(G, S)$ is an equilibrium solution iff $\alpha_2 > \beta + \delta$. In this case $(S, G)$ is also an equilibrium because $\alpha_1 > \alpha_2 > \beta + \delta$ and $r_2(S, G) > r_2(S, S)$. But if $\alpha_2 < \beta + \delta$ then $r_2(G, G) > r_2(G, S)$ and again there is a single unique equilibrium determined by whether $\alpha_1 > \beta + \delta$. $\qquad \square$

**Theorem C.2.** *Consider an n-player version of the game from Theorem 1. If there exists some $i$ such that $\alpha_i > \max_{j \neq i} \alpha_j + n\delta$ and $\alpha_i > \beta + \delta$, then the game has a unique equilibrium $\{Z_i = S, Z_{j \neq i} = G\}$. If $\alpha_i < \beta + \delta$ for all $i$ then there is a unique equilibrium $\{Z_i = G\}$.*

*Proof.* Let $r_i(Z)$ indicate the reward for agent $i$ from the strategy vector $Z = (Z_1, Z_2, \ldots, Z_n)$. Let $\overline{\alpha}(Z) = \frac{1}{\sum_i 1[Z_i = S]} \sum_i 1[Z_i = S]\alpha_i$. We will use the shorthand $n_{-i} = \sum_{j \neq i} 1[Z_j = S]$ and

$$\overline{\alpha}_{-i} = \begin{cases} \frac{1}{n_{-i}} \sum_{j \neq i} 1[Z_j = S]\alpha_j, & \exists_{j \neq i} Z_j = S \\ \beta, & \text{otherwise.} \end{cases} \tag{1}$$

Note that $r_i(Z_{-i} \oplus Z_i := G) = \overline{\alpha}(Z_{-i} \oplus Z_i := G) = \overline{\alpha}_{-i}$.

First consider the case $\max_{j \neq i} \alpha_j > \beta$.

$$\alpha_i > \max_{j \neq i} \alpha_j + n\delta \tag{2}$$

$$\geq \overline{\alpha}_{-i} + (n_{-i} + 1)\delta \tag{3}$$

$$n_{-i}\overline{\alpha}_{-i} + \alpha_i > \overline{\alpha}_{-i} + (n_{-i} + 1)\delta + n_{-i}\overline{\alpha}_{-i} \tag{4}$$

$$n_{-i}\overline{\alpha}_{-i} + \alpha_i > (n_{-i} + 1)\overline{\alpha}_{-i} + (n_{-i} + 1)\delta \tag{5}$$

$$n_{-i}\overline{\alpha}_{-i} + \alpha_i - (n_{-i} + 1)\delta > (n_{-i} + 1)\overline{\alpha}_{-i} \tag{6}$$

$$\frac{1}{(n_{-i} + 1)}(n_{-i}\overline{\alpha}_{-i} + \alpha_i) - \delta > \overline{\alpha}_{-i} \tag{7}$$

$$\overline{\alpha}(Z_{-i} \oplus Z_i = S) - \delta > \overline{\alpha}_{-i} \tag{8}$$

$$r_i(Z_{-i} \oplus Z_i = S) > r_i(Z_{-i} \oplus Z_i = G). \tag{9}$$

Thus, when $\alpha_i > \max_{j \neq i} \alpha_j + n\delta > \beta + n\delta$, player $i$ cannot improve its reward by deviating from $Z_i = S$, regardless of what the other players do. Similarly, as long as the $\alpha$-maximizing player bids $Z_i = S$, the $S$-bidding player with smallest $\alpha_j$ can improve its reward by switching to $Z_j = G$, since this will increase $\overline{\alpha}(Z)$ and also avoid the penalty $\delta$. In equilibrium $Z_j = G$ for all $j \neq i$.

Now consider the case when $\max_{j \neq i} \alpha_j < \beta$. Again, among the players who play $Z_j = S$, the one with the smallest $\alpha_j$ can always improve its reward by switching to $Z_j = G$, which increases $\overline{\alpha}$ and decreases its effort penalty. Ultimately then all $Z_{j \neq i} = G$. If $\alpha_i > \beta + \delta$ then $Z_i = S$, resulting in the same unique equilibrium $\{Z_i = S, Z_{j \neq i} = G\}$. If $\max_i \alpha_i < \beta + \delta$ then no player can improve their reward by deviating from $Z_j = G$. $\qquad \square$

## D Equilibria of Table 1

We show the pure strategy equilibria of the game introduced in Table 1 as $\alpha_1$ and $\alpha_2$ vary.

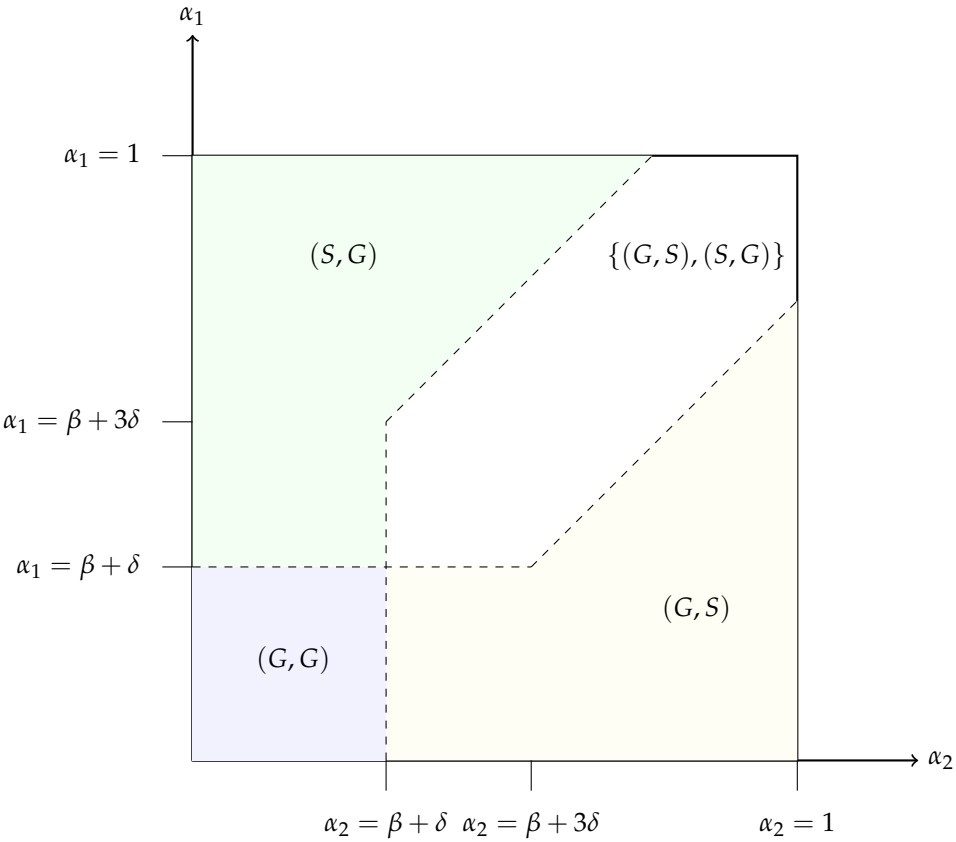

Figure 6: Pure strategy equilibria of the game introduced in Table 1 as $\alpha_1$ and $\alpha_2$ vary. Note that there are multiple equilibria when $\alpha_1$ and $\alpha_2$ are sufficiently high and sufficiently similar.

# E  ReST learning dynamics

Here, we report all metrics (task performance, search rate, search calibration, and answer calibration) for all models after each ReST epoch. The search rate decreases monotonically over the first two epochs, and in most cases continues to decrease in the third epoch (Figure 8). Meanwhile task performance increases significantly for the "mismatched" settings (pubmed/popqa, wiki/bioasq) while decreasing slightly for the "matched" settings (pubmed/bioasq, wiki/popqa), see Figure 7. These two findings are consistent because we already observed that the retrieval results slightly impair performance in the mismatched settings; by searching less frequently, the agents trade off slight regressions in the matched settings for large improvements in the mismatched settings, and a lower effort penalty.

Figure 9 measures search calibration across ReST epochs by showing the average $F_1$ when sorting questions by P(#SEARCH), and using #SEARCH in 10%/20%/50% of the questions (the setup introduced in § 6). Search calibration increases in the first two ReST epochs for all methods, plateauing in the third epoch. Similarly, Figure 10 measures answer calibration across ReST epochs by showing average $F_1$ when sorting questions by P(#ANSWER) and answering only 10%/20%/50% of the questions (the setup introduced in § 6). The calibration of $P(\text{\#ANSWER})$ improves over the first two epochs for collaborative self-play (CSP), but is much less stable for the deanonymization ablation (CSP-DeAnon). We hypothesize this is because the asker learns to attend to the helper identity in the first epoch, which results in the #ANSWER vs. #GUESS choice being less correlated with reward in the rollouts that comprise the training data for the second epoch.

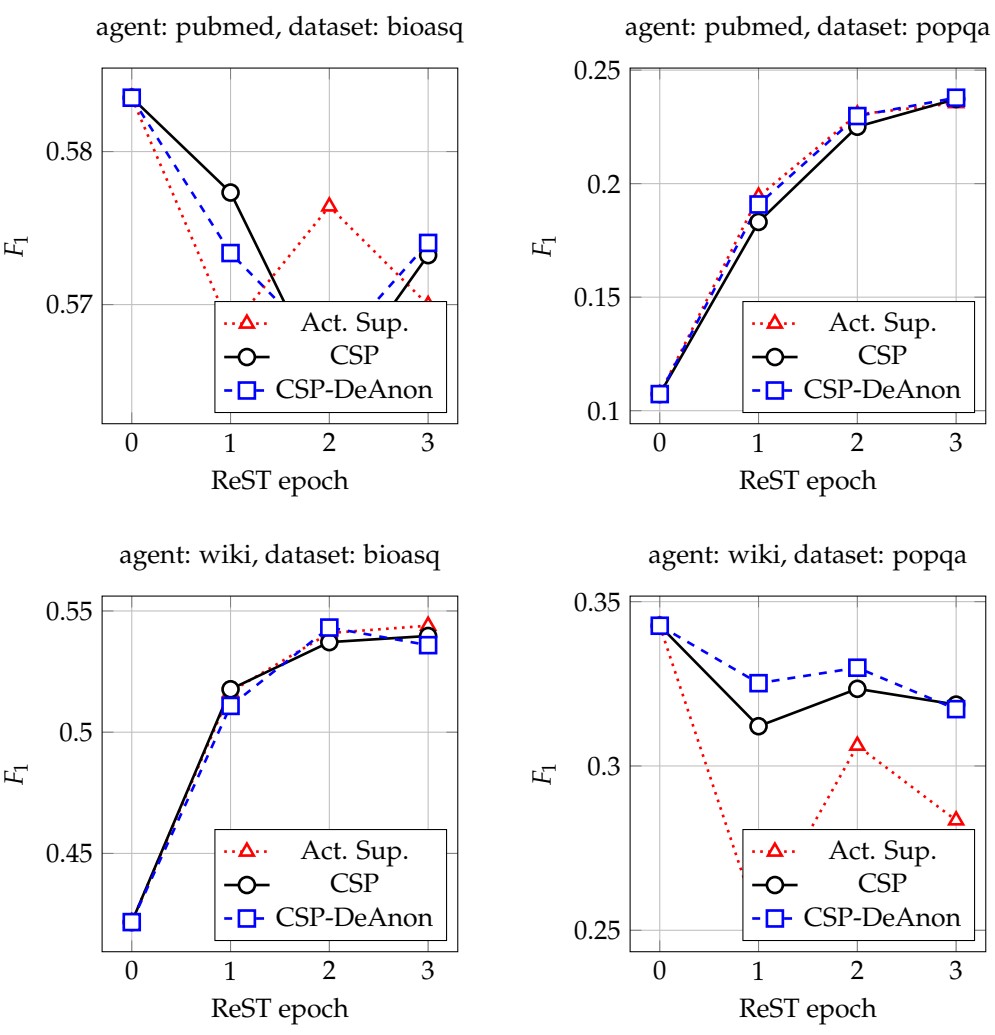

Figure 7: Task performance per epoch of ReST

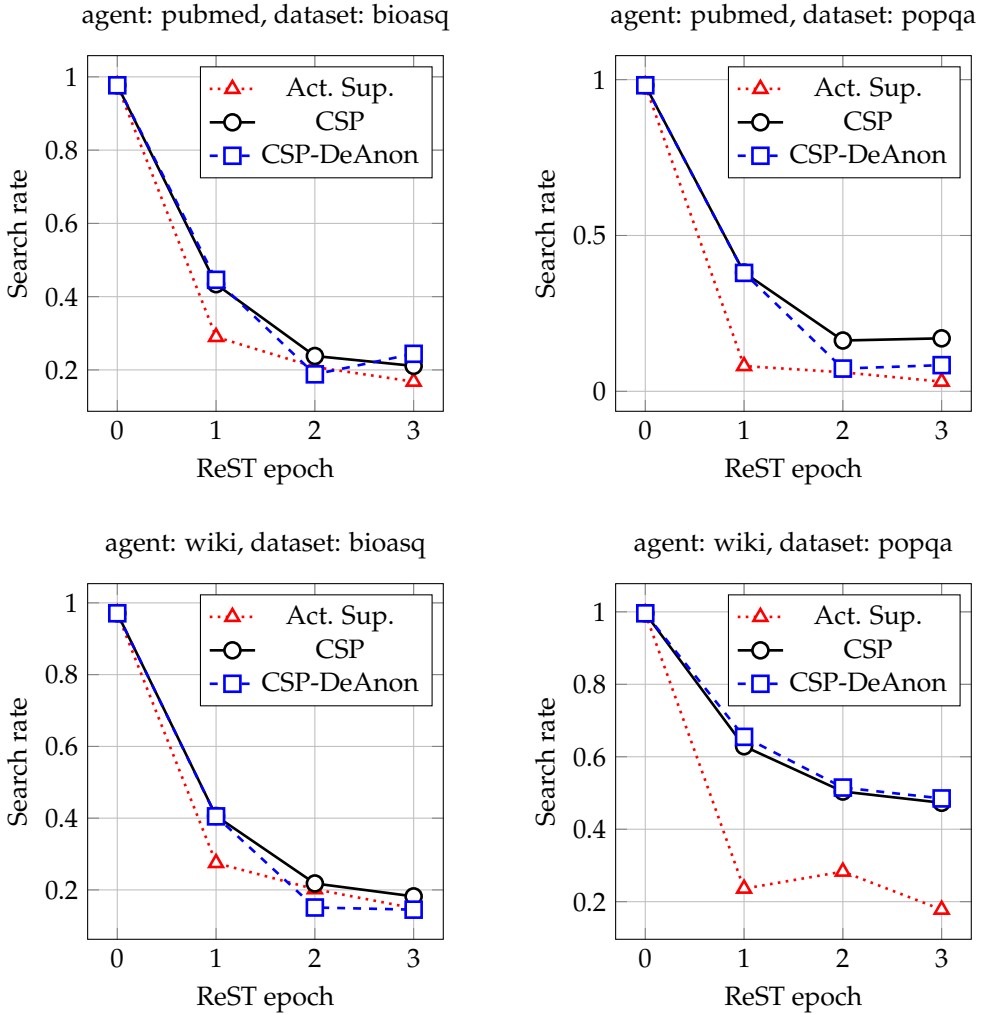

Figure 8: Search rate per epoch of ReST

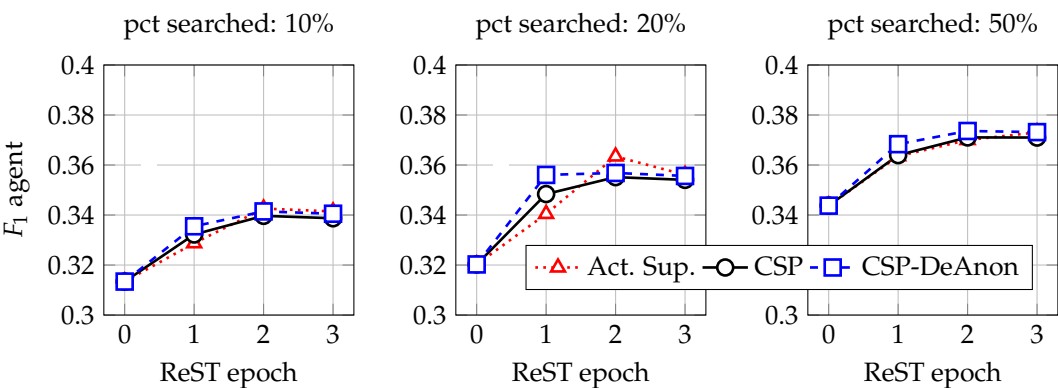

Figure 9: **Calibration of P(#SEARCH) through ReST**. Same setting as Figure 4. We show average $F_1$ when issuing search only for 10%/20%/50% of the queries with highest P(#SEARCH) for all ReST epochs. Calibration improves in the first and second ReST epochs (evidenced by higher $F_1$ when using #SEARCH in only a fraction of the questions), and is similar across methods.

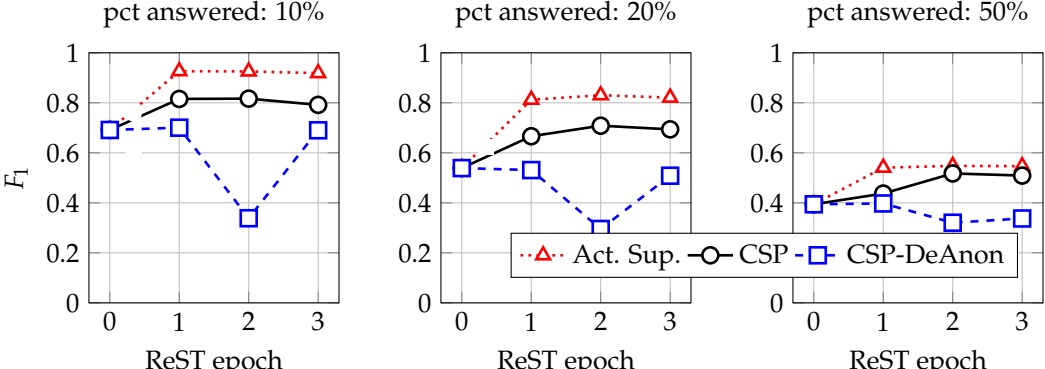

Figure 10: **Calibration of P(#ANSWER) through ReST**. Same setting as Figure 5. We show average $F_1$ when answering 10%/20%/50% of the questions with highest P(#ANSWER). Answer calibration improves (evidenced by higher $F_1$ when answering only a fraction of the questions) for action supervision (Act. Sup.) and collaborative self-play (CSP), but not for CSP-DeAnon, because the asker can achieve high task accuracy without attending to the helpers' confidence.

