# OpenReview forum: "Don’t lie to your friends: Learning what you know from collaborative self-play"
_colmweb.org/COLM/2025/Conference — COLM 2025_

### Official Review · Reviewer_osPm · 2025-05-12

**Rating:** 8
**Confidence:** 4
**Ethics Flag:** 1

**Summary:**

This paper considers the problem of teaching AI agents to be aware of their own knowledge capabilities and limitations. It proposes to use collaborative self-play (CSP; multiple copies of the same language agent working together in a controlled environment to generate a training signal for that same agent) as a framework for training these skills in language agents. Experiments in the setting of question answering using external tools (reliable knowledge sources) demonstrate the framework and provide compelling evidence for its effectiveness.

**Questions To Authors:**

* The nature of the orchestrator is not clear. Is this another language model? The same model as the self-play-ers? A hard-coded policy?
 * 'If the agent posers a question,...' (Ln 113) how is this determined? Regular expression parsing? Are you looking for certain literal token sequences (like `#ASK`?)
 * What is the middle circle operator on Line 146?
 * It seems like the most important immediate direction the current work could be extended is to add search-then-hedge and guess-then-answer actions - this would also teach the models when to trust their tools or their internal knowledge, right? Have you tried such experiments?

**Reasons To Accept:**

* The proposed framework is simple and flexible - it could readily be applied to other aspects beyond question answering with knowledge tools.
 * The paper is clearly written and easy to read
 * The problem studied is very important, relevant to COLM, and timely (c.f. OpenAI issues with the related problem of sycophancy in the news last week).
 * The empirical results for the presented case study are strong, and the analysis covers a range of relevant aspects
 * The proposed training framework uses natural language as an intermediary, offering a degree of interpretability to the training process

**Reasons To Reject:**

* The paper only presents results for one setting - question answering with knowledge retrieval tools
 * The use of self-play for generating a training signal is not new - not even for language agents - e.g. consider alpha zero, or the thread of work using Hanabi as a multi-agent benchmark for LLM collaboration and reasoning. This body of literature is completely missing in the discussion of related work. The paper needs to engage with this prior work to provide better context for the novelty of the current contribution.
 * The discussion of past studies on LLM confidence calibration is very brief. How does this paper shed new light on e.g. previous results that LLMs seem to have a degree of self-calibration in their own answer confidence? E.g. see Tian for a jumping off point in the literature [1]
 * Some details of the framework are not clearly explained (see Q's to authors below).
 * The CSP framework (Alg. 1) seems like it would be (very) inference intensive. A token level complexity analysis or even empirical results describing the token budget or compute cost scaling is missing from the paper, but is needed to provide a wholistic picture of the effectiveness of the method.
 * The current paper only considers one LLM - gemma2-9B (I believe).

### References
[1] Tian, Katherine, et al. "Just ask for calibration: Strategies for eliciting calibrated confidence scores from language models fine-tuned with human feedback." arXiv preprint arXiv:2305.14975 (2023).

---

> ### Author Response · Authors · 2025-06-02
> **prior work on self-play and calibration; analysis of inference costs; clarification of the orchestration algorithm**
>
> Thanks for your careful review and thoughtful questions!
>
> - **Prior work on self-play training and confidence calibration**
>
> Thanks for raising this point. We plan to expand and restructure the discussion of related work, and will add a subsection on prior work on self-play training. The main contrast with work such as the Hanabi challenge is that our focus is not on learning to play cooperative games per se, but specifically on using collaborative games to be better collaborative helpers. For this reason, we construct a game that connects much more closely with real information-seeking queries, demonstrating that self-play can help train agents in this practically-relevant user-facing setting, not just training agents to play games with other agents.
>
> We will also add a discussion on prior work on calibration, including Tian et al (2023). Most of this work focuses on calibration with respect to parametric knowledge and not retrievals. The ICL method shown in Figure 5 is somewhat similar to the "Ling-1s" method of Tian et al, and we observe relatively poor calibration in the sense of correlation between P(Answer) and F1 of the parametric model. See also our response to reviewer GEzH, in which we compute the correlation between P(Answer) and parametric F1 as .2-.3).
>
> - **Inference costs**
>
> Inference is not a significant cost because the outputs are short. Specifically, generating the training data required 64K rollouts per round of ReST training (1000 queries per dataset, 32 rollouts per query), with 3-5 inference calls per rollout. This yields an upper bound of 1.28M inference calls, each of which resulted in 5-10 tokens. Examination of our logs show that a single ReST epoch produced less than 2M whitespace-delimited tokens, so the entire CSP training budget is <10M output tokens. At third-party prices for gemma2-9b (https://groq.com/pricing/), this would cost less than $2 US. We will add these details to the next version of the paper.
>
>
> - **Q1 & Q2: orchestration**
>
> Section 3 describes the setup at a very abstract level, which could apply to many possible orchestration setups. As described in section 5, the orchestrator always passes the question to the helpers in our experiments. Extension to more flexible orchestration policies is left to future work, but this is orthogonal to this paper's main focus, which is on training the helpers to be well-calibrated and efficient tool users.
>
> - Q3: (Line 146)
>
> The middle circle represents concatenation.
>
> - Q4: **It seems like the most important immediate direction the current work could be extended is to add search-then-hedge and guess-then-answer actions - this would also teach the models when to trust their tools or their internal knowledge, right?**
>
> Can you say more about this proposal? By decomposing search/guess and hedge/answer into separate actions, the agents can learn to hedge when the search results are disappointing. If the agent knew in advance that search was not going to yield good results, it could hedge without searching and thereby avoid paying for the tool call.

---

> > ### Comment · Reviewer_osPm · 2025-06-04
> > **Response to authors**
> >
> > Thank you for the detailed response here and in the comment to all authors. Your comments have clarified some of my confusion about a few points. I'll leave my review scores as-is.
> >
> > RE: Expanding the methodology: "it could hedge without searching and thereby avoid paying for the tool call." - I think this is what I was getting at - having the agent learn to trade off the value of (potential) information gained through tool calls is a valuable next step for this research agenda I believe.
> >
> > Best of luck with preparing the updated version of the paper!

---

### Official Review · Reviewer_T7Se · 2025-05-12

**Rating:** 7
**Confidence:** 3
**Ethics Flag:** 1

**Summary:**

This paper introduces collaborative self-play (CSP), a novel framework for training language models to improve tool usage and uncertainty calibration through multi-agent interactions. The core idea is to design a multi-agent environment where agents with heterogeneous tools (e.g., corpus-specific retrieval) collaborate to answer questions, incentivizing them to learn when to trust their parametric knowledge, use tools, or hedge. Empirical validation on factoid QA benchmarks demonstrates the effectiveness of this paper’s idea.

**Questions To Authors:**

1. In the implementation, the Reinforced Self-Training (ReST) method used by the authors is essentially a form of Reject Fine-tuning. Have you tried other reinforcement learning (RL) algorithms? If so, what issues do you think might arise from using alternative RL approaches in this framework?
2. The layout of the article's appendix is suggested to be adjusted, as there is currently much whitespace.

**Reasons To Accept:**

1. **Novel Training Paradigm**: This paper introduces a novel training paradigm by leveraging multi-agent collaboration, which seems to be a promising approach compared to traditional supervised fine-tuning methods.

2. **Effective Empirical Results**: The empirical experimental results are good, and the conclusions are reasonable.

3. **Good Theoretical Analysis**: Theoretical analysis is provided on an information-provision game in this paper. Although this a more simplified and limited scenario, this part of theoretical analysis contributes to the overall completeness of the work.

**Reasons To Reject:**

1. While the collaborative self-play framework is conceptually novel, its implementation and validation are narrow and simplistic. The generalization to other domains or more complex interaction is leaved unproven. Besides, the study only examines small agent numbers, ignoring possible optimization challenges that arise in larger multi-agent scenarios.

---

> ### Author Response · Authors · 2025-06-02
> **next steps**
>
> Thanks for your review, which highlights several interesting possibilities for future work.
>
> - **"The generalization to other domains or more complex interaction is leaved unproven. Besides, the study only examines small agent numbers..."**
>
> We choose a relatively focused application of collaborative self-play to isolate the key scientific questions involved in learning calibration and tool use. Scaling to other domains and interaction patterns may teach other skills, such as planning and delegation. We are excited to pursue these possibilities in future work. Similarly, extension to larger agent societies is an exciting direction, though as we show, even simple three-agent societies are sufficient for interesting and productive capabilities to emerge.
>
> - **"Have you tried other reinforcement learning (RL) algorithms? If so, what issues do you think might arise?"**
>
> We are interested to explore the applicability of other RL algorithms, but see at least two potential issues. First, these algorithms will be less efficient than ReST, which benefits by interleaving easily-parallelizable inference steps with parameter updates. Second, on-policy algorithms might be more able to hack the reward, in the sense of learning strategies that work well in the training setting but transfer poorly to interactions with human users.

---

> > ### Comment · Reviewer_T7Se · 2025-06-05
> >
> > Thank you for your careful response and the clarifications provided to all reviewers. I have reviewed the authors' rebuttal and the other reviewers' comments. I will maintain my original rating and look forward to seeing the future development of this work.

---

### Official Review · Reviewer_GEzH · 2025-05-12

**Rating:** 7
**Confidence:** 4
**Ethics Flag:** 1

**Summary:**

This paper introduces a multi-agent training framework to improve the calibration and tool-use capabilities of language models.

The experiment setup is a question-answering task in which N=3 agents (each sharing the same parameters but different prompts) must communicate and use retrieval tools in order to answer questions. One of these agents is given the initial question, and it can submit queries to the other two agents, each of which have access to their own, domain-specific retrieval tools (one for Wikipedia, and one for PubMed). Given a question from the first agent, the two retrieval agents can conduct search in their respective domains and then return answers, conditioned either on the results of their queries or on knowledge stored in their parameters (which are the same across agents), which are sent to the initial agent (along with a binary expression of certainty)

The models are trained on high-scoring interactions (using a slightly modified version of ReST, i.e., iterated filtered BC). This leads the models to produce indicators of uncertainty when generating answers based on their parametric knowledge, rather than on retrieved documents. Additionally, when given a small penalty for each tool use, the model learns to only retrieve external sources when necessary.

**Questions To Authors:**

1. Does the finding that models learn to use the retrieval tool less frequently over the course of training require a multi-agent setup? I'd think you could show a similar result in a single agent setting (but perhaps I'm missing something here)

In case it's useful, here are some potential additional citations on self-play training + emergent behaviors:
[1] Wang, R., Yu, H., Zhang, W., Qi, Z., Sap, M., Bisk, Y., ... & Zhu, H. (2024, August). SOTOPIA-π: Interactive Learning of Socially Intelligent Language Agents. In Proceedings of the 62nd Annual Meeting of the Association for Computational Linguistics (Volume 1: Long Papers) (pp. 12912-12940).
[2] Liao, A., Tomlin, N., & Klein, D. (2024). Efficacy of language model self-play in non-zero-sum games. arXiv preprint arXiv:2406.18872.

**Reasons To Accept:**

1. The central idea of this paper is that expressions of uncertainty emerge due to their effect on the behavior of downstream listeners. This approach feels fundamentally correct, and I think scaling it up might be able to fix linguistic calibration in LLMs more broadly. In general, I really appreciate the approach of showing how desirable properties like calibration can emerge through communicative incentives

2. The experiment design is straightforward and clear. I think having three agents (two "speaker" agents and one "listener" agent) is the minimal setting needed to observe this type of calibration behavior, and using two different retrieval datasets is a clean + simple approach that could reasonably be reused in followup work

3. In general, the paper is well-written and well-motivated

**Reasons To Reject:**

1. While I like the simple evaluation setup, I find some parts to be oversimplified. In particular, the use of the special tokens #HEDGE and #ANSWER instead of just letting models generate freeform expressions of uncertainty seems unrealistic and limits the usefulness of the trained models. It's also unclear to me why the paper made this simplification -- are there other challenges needed to scale this up to real linguistic calibration?

2. I found the presentation of the results section to be a bit confusing at first. In particular, the "search calibration" and "answer calibration" results aren't what I would have initially expected given their titles, i.e., I was expecting to see some calibration measure, e.g., ECE. In order to make this section more clear, I'd recommend first describing the expected output for a calibrated system and then showing how your approach matches this

---

> ### Author Response · Authors · 2025-06-02
> **freeform outputs combining answer & confidence; quantifying calibration; comparison to other collaborative self-play games**
>
> Thanks for the thoughtful review! We have conducted some additional experiments to answer your questions and concerns.
>
> - **"[T]he use of the special tokens #HEDGE and #ANSWER instead of just letting models generate freeform expressions of uncertainty seems unrealistic and limits usefulness"**
>
> It is relatively easy to go from this structured representation to freeform text. As a pilot experiment, we prompted gemma2-9b to go from (question, answer, confidence) tuples to freeform generations. To increase syntactic diversity, we then asked gemma2 to paraphrase each generation zero, one, or two times. Finally, to test that the freeform generations accurately show the confidence, we then prompted gemma2-9b to predict the confidence from the free-form text. The round-trip confidence consistency on 100 examples was 91%, with all but one of the errors being a GUESS mislabeled as an ANSWER, due to the paraphrase prompt introducing overconfidence. Here are some examples of the freeform outputs:
>
> ['Terrence McNally penned the captivating play, Sweet Eros.' (ANSWER),
>  "While I can't guarantee it, I am very confident that Marvejols is located in France." (GUESS),
>  'I strongly suspect, though not conclusively, that burning mouth syndrome affects post-menopausal women more frequently.' (GUESS),
>  'John le Carré is the creative force responsible for the novel The Dancer Upstairs.' (ANSWER),
>  'All evidence points to David Lynch as the probable producer of BE.' (GUESS)]
>
> This experiment suggests that our approach could be applied to free-text communication between the agents, and could provide the user with free-text representations that combine the confidence and the answer.
>
> - **"I was expecting to see some calibration measure, e.g., ECE."**
>
> Thanks for this comment, we agree this would be helpful. We have computed the Spearman correlation between P(answer) and the F1 of the prompted model (parametric knowledge), which corresponds to a statistical summary of the relationship shown in Figure 5. Rank correlation is more appropriate than ECE or Brier for this case, because the evaluation metric is F1 and not accuracy.
>
> In-distribution (popqa, bioasq)
>
> | | setting | spearman |
> |---:|:--------------------|----------:|
> | 0 | CSP 3 | 0.536258 |
> | 1 | CSP-deanon 3 | 0.0695959 |
> | 2 | action supervision 3 | 0.663741 |
> | 3 | prompted | 0.231374 |
>
> Out-of-distribution (nq, entq)
>
> | | setting | spearman |
> |---:|:--------------------|----------:|
> | 0 | CSP 3 | 0.366328 |
> | 1 | CSP-deanon 3 | 0.167744 |
> | 2 | action supervision 3 | 0.39914 |
> | 3 | prompted | 0.317859 |
>
> By the Fisher transform all correlations are statistically significant at p<.05 (z=2.1 for the correlation \rho=.069). CSP and action supervision significantly improve on in-context learning, and as discussed in the text, deanonymization dramatically reduces the calibration of CSP.
>
> - **Prior work**
>
> Thanks for these pointers. We plan to add significantly more discussion of prior work in the paper, including these two references. A key contrast is that our focus is not on learning to play cooperative games per se, but specifically on using collaborative games to be better collaborative helpers. For this reason we design a new game based around realistic user information-seeking queries, rather than automatically generated social tasks (Sotopia) or negotiation games (Tomlin et al).
>
> - **"Does the finding that models learn to use the retrieval tool less frequently over the course of training require a multi-agent setup?"**
>
> The retrieval rate decreases across rest epochs for both multi-agent and single-agent training (CSP and Action Supervision; see figure 8 in the appendix). This is likely an artifact of the very high search rate of the prompted model.
>
> We hope the reviewer will consider raising their score given the new results provided, explanations and commitment to improve the discussion of related work.

---

> > ### Comment · Reviewer_GEzH · 2025-06-04
> >
> > Thanks for your response! At the moment, I'd like to keep my score as-is, primarily due to the small scale/simplified experimental setup in the paper. That said, I still think this is good work and recommend it for acceptance!
> >
> > To followup on one point:
> >
> > > #ANSWER and #HEDGE
> >
> > I agree that today's language models are totally capable of mapping from the structured (answer, confidence) representation to natural language. To clarify: my main concern here isn't whether this mapping is possible, but whether the confidence representation is sufficiently expressive to capture the full range of possible expressions of uncertainty. In other words, if you use the #ANSWER and #HEDGE tokens as a bottleneck, then you can only generate binary expressions of uncertainty, e.g., including a phrase like "while I can't guarantee it" in the #HEDGE case.
> >
> > However, in natural language, uncertainty can be (1) graded, and (2) explained via additional factors. To use one of your examples, we might want models that can generate sentences like, "I've seen some studies that suggest that that burning mouth syndrome affects post-menopausal women more frequently, but I haven't seen newer data, and I'm unsure if there is an established scientific conclusion" (where the source/type of uncertainty is specified). To get this type of language to emerge from a communicative objective, I believe you'd need to move beyond the #ANSWER/#HEDGE setup.

---

### Official Review · Reviewer_VkVL · 2025-05-13

**Rating:** 7
**Confidence:** 4
**Ethics Flag:** 1

**Summary:**

The paper discusses a collaborative self-play approach to (1) guide LLMs to differentiate between parametric knowledge and tool usage (2) when to trust the tool o/p (calibrate the response of tool output) (3) when to abstain / hedge. The use-case discussed is a q&a setup with multiple agents having access to diverse knowledge bases, and one want to tradeoff cost (of API calls) with accuracy. It frames the problems as multi-agent so that the proper strategy to use tools emerges from this multi-agent setup (that share parameters) during training and translated to single-agent setup during inference.

**Questions To Authors:**

1. There is a body of work around of aleotoric vs epistemic uncertainty in LLMs , see this for eg: https://arxiv.org/pdf/2406.02543v1. I think this needs to be added to references and discussed in the context of the use case of the paper, as it is very relevant

2. The idea of self-play and using game theory for dialog has been the subject of this study in the context of goal oriented dialogs: https://arxiv.org/abs/2109.09597. Again this is relevant background work as some of the ideas find a mention in the linked study

3. [minor] Fig 2 needs to be annotated with (a, v, t) labels, to make it more readable

4. Section 3: In this paper: https://papers.nips.cc/paper_files/paper/2017/file/68053af2923e00204c3ca7c6a3150cf7-Paper.pdf the authors   mention the problem of confirmation bias in DNNs. Why does the multi-agent approach discussed in the paper not lead to this confirmation bias problem ?

5. The hanabi challenge https://arxiv.org/abs/1902.00506 discusses another co-operative game. It might be useful to compare the present work to this cooperative game and explain similarities and differences

6. [minor] Table 2 needs to be annotated more with important cells highlighted for easy readability.

7. Is there a skyline in Table 2 or is action supervision that skyline ? In otherwords, can we design a scenario where an oracle tells whether to search vs hedge perfectly, and what would the performance be ?

8. What is the advantage of CSP ie the approach presented in this work vis-a-vis Action Supervision ? (results section)

9. [minor] line 298: why is the search rate so low compared to action supervision ?

**Reasons To Accept:**

The paper is lucid, and motivated well. It also contains an in-depth experimental setup. The use-case (although is a very specific one) make intuitive sense, as it is a key question that is critical to LLM nowadays where it is crucial for the LLM determine when to use parametric knowledge vs tools, when to hedge and abstain from answering and when to answer a user request, if the LLM can be sure of its response, without hallucinating.

**Reasons To Reject:**

At a high level, I do not see any major limitations of the presented paper. One minor issue I see is that the presented approach has little performance improvements compared to the baselines. I do not see a clear explanation for this narrow performance improvement.

I do have several questions to the authors (see below). If those can be addressed, the paper can further strengthened.

---

> ### Author Response · Authors · 2025-06-02
> **missing references and eval discussion**
>
> Thanks for the thoughtful review and interesting questions! Responses are inline.
>
> - **"narrow performance improvements compared to the baselines" "Q8. What is the advantage of CSP vs action supervision?"**
>
> Empirically, we find that action supervision excels at training an agent to select correct actions within a given task, but is less effective at learning a policy that transfers well to new settings, as demonstrated on the heldout datasets EntQ and NQ, where it has relatively weak F1 (table 2) and poor search calibration (figure 4b).
>
> More importantly, action supervision relies on heuristically determining the optimal action at each step of training. This will not be possible in every setting and is strictly less general than collaborative self-play training from the group reward alone. For example, it is unclear how to do action supervision in a multi-hop setting in which distinct information is needed from each agent. Even in our setting, action supervision requires an arbitrary F1 threshold, which is not required by collaborative self play.
>
> Stepping back, CSP can be viewed as a form of outcome supervision while action supervision is process supervision. In general, we can obtain outcome supervision on a broader set of problems and techniques than those on which it is possible to obtain process supervision.
>
> - **Q1. aleatoric vs epistemic uncertainty**
>
> Thanks for raising this point. We plan to expand and restructure the discussion of related work to cover uncertainty quantification more comprehensively, including aleatoric vs epistemic uncertainty. From the perspective of the helper agents, the retrieval tool can reduce epistemic uncertainty about questions that are not in the model's parametric knowledge; from the asker's perspective, confidence markers can reduce epistemic uncertainty about which helper agent to trust.
>
> - **Q2. prior work on game theory for dialog**
>
> Thanks again, we acknowledge the related work would benefit from a deeper discussion of this literature, which we plan to add in the next version. We will include the suggested reference as well as references therein, particularly the line of research from Prashant Parikh.
>
> - **Q3. Fig 2 needs to be annotated with (a, v, t) labels, to make it more readable**
>
> Sorry but we do not understand the suggestion. Can you be more concrete?
>
> - **Q4. Confirmation bias**
>
> This comment references a paper on semi-supervised learning, in which the teacher generates answers that may be incorrect. In our setting all answers are grounded in gold references. Could you please say more about how this issue may relate to our paper?
>
> - **Q5. Hanabi challenge**
>
> Thanks for this reference. The Hanabi challenge paper does share our interest in learning to play collaborative games. However, we emphasize that our ultimate goal is not on learning to play cooperative games per se, but specifically on using collaborative games to train LMs to become better collaborative helpers. For this reason we design a new game based around realistic user information-seeking queries. We'll be sure to include this motivation in the next version of the paper.
>
> - Q6. [minor] Table 2 needs to be annotated more with important cells highlighted for easy readability.
>
> Thanks for the suggestion, we will restructure this table to improve readability in the next version.

---

> > ### Author Response · Authors · 2025-06-02
> > **skyline results and search rate discussion**
> >
> > - Q7. **"Is there a skyline in Table 2** or is action supervision that skyline? In other words, can we design a scenario where an oracle tells whether to search vs hedge perfectly"
> >
> > At training time, action supervision has access to such an oracle. We can compute this skyline post hoc at test time, using the RAG result in exactly those cases where it improves F1 over a parametric answer:
> >
> > |                                           |   oracle_f1 |   oracle_search_rate |
> > |:------------------------------------------|------------:|---------------------:|
> > | ('bioasq', 'pubmed_gecko_agent', 'CSP 3') |    0.690049 |                0.294 |
> > | ('bioasq', 'wiki_bm25_agent', 'CSP 3')    |    0.605748 |                0.175 |
> > | ('popqa', 'pubmed_gecko_agent', 'CSP 3')  |    0.256208 |                0.048 |
> > | ('popqa', 'wiki_bm25_agent', 'CSP 3')     |    0.427151 |                0.248 |
> > | ('entq', 'pubmed_gecko_agent', 'CSP 3')   |    0.383779 |                0.075 |
> > | ('entq', 'wiki_bm25_agent', 'CSP 3')      |    0.618937 |                0.379 |
> > | ('nq', 'pubmed_gecko_agent', 'CSP 3')     |    0.42819  |                0.079 |
> > | ('nq', 'wiki_bm25_agent', 'CSP 3')        |    0.536772 |                0.209 |
> >
> > Overall, there is .1-.2 points of headroom in F1 from the oracle. Search rates are generally comparable, with the pubmed tool helping less frequently on the non-bioasq datasets.
> >
> > - Q9. **"why is search rate so low compared to action supervision"**
> >
> > Line 298 notes that action supervision yields a very low search rate, which is based on the frequency with which search improves performance in the wiki/popqa setting. On EntQ, more searching would be helpful, but action supervision has apparently somewhat overfit the properties of the PopQA training set. Collaborative Self Play more effectively trains the model to use search when it is likely to yield improvements.
> >
> > We hope the reviewer will consider raising their score given the above answers, the new results, and the commitment to improve clarity w.r.t prior work.

---

> > ### Comment · Reviewer_VkVL · 2025-06-09
> > **Response to author comments on review feedback**
> >
> > Thanks for incorporating some of the suggestions to make the paper solid.
> >
> > My point about the figure was that it missed some annotation such as what is the strategy used by the answering agent in terms of the notation mentioned in the equations and narration later.
> >
> > About the confirmation bias - perhaps your setting is different and might not lead to this bias problem. Feel free to ignore this suggestion if it does not apply, if not a short line or two as to clarify it will help. I forget the details of the approach, but when I was reading the paper I got this question in my mind ..

---

### Author Response · Authors · 2025-06-02
**thanks for your reviews!**

Thanks to all four reviewers for their thoughtful feedback on our submission! We are glad that their assessments are positive, finding the paper "lucid" and "well-motivated", with an "in-depth experimental setup", "strong" empirical results, and a "good theoretical analysis." The reviewers also make several valuable suggestions for improving the work:

1. **Better literature review**, including prior work on uncertainty (aleatoric vs epistemic, verbalized uncertainty), game theory for dialogue, and prior work on learning from collaborative games, such as the Hanabi challenge and Sotopia-pi. We truly appreciate these sections and will substantially expand and restructure the related work in the next version. Specifically regarding prior work on learning from collaborative games, a key difference is that we focus on using multi-LM collaboration to improve capabilities that will be used even in single-LM interactions with humans, in contrast to work like the Hanabi challenge, where multi-LM collaboration is the end goal.
1. **Better quantification of calibration**: in the response to GEzH we offer new results showing the rank correlation between P(Answer) and F1, quantifying the relationship shown visually in Figure 5.
1. **Integrated certainty markers and answers**: we perform a pilot study of the ability of gemma to integrate certainty markers and answers post-hoc, demonstrating a high rate of round-trip consistency. Thus, if this form of output is desired, it can be obtained relatively easily by prompting.
1. **Analysis of inference costs**. In the response to osPm, we estimate the total inference costs for collaborative self-play using a third-party API provider. Although we issue many queries, the cost is low (<$10 US) because the outputs are short.
1. **Skyline results**: reviewer VkVL asks about the "skyline" cost/accuracy tradeoff, assuming search is used only when it helps. We compute these numbers, showing that there is still headroom in terms of average F1 at the same or lower retrieval rates.

We again thank the reviewers for these suggestions, and look forward to incorporating all of these additional findings and references into the next version of the paper. If this addresses any of your concerns, we hope you will consider raising your scores.

---

### Decision · Program_Chairs · 2025-07-08

**Decision:**

Accept

**Comment:**

This paper studies how to train LLMs to (1) choose between relying on parametric knowledge or external tools, (2) decide whether to trust the outputs of these tools, and (3) determine when to abstain or hedge. The authors propose a novel training paradigm based on collaborative self-play, where one agent receives the question and, together with two other agents—each with access to a different external corpus—collaborates to provide answers.

The reviewers (VkVL, GEzH, osPm) generally agree that the paper is well-written and well-motivated. Reviewer T7Se highlights the novelty of using collaborative self-play to improve tool usage and uncertainty calibration. The reviewers also find the empirical results and analysis to be convincing (VkVL, GEzH, T7Se, osPm).

Similar to all the reviewers, I personally also like the idea of this paper to leverage the power of multi-agent collaboration. I would be more than happy to express my support for this work.